# A Unified Hard-Constraint Framework for Solving Geometrically Complex PDEs

**Songming Liu**[1], **Zhongkai Hao**[1], **Chengyang Ying**[1], **Hang Su**[1,2]*, **Jun Zhu**[1,2]*, **Ze Cheng**[3]

[1]Dept. of Comp. Sci. and Tech., Institute for AI, THBI Lab, BNRist Center,
Tsinghua-Bosch Joint ML Center, Tsinghua University
[2]Peng Cheng Laboratory; Pazhou Laboratory (Huangpu), Guangzhou, China
[3]Bosch Center for Artificial Intelligence
csuastt@gmail.com

## Abstract

We present a unified hard-constraint framework for solving geometrically complex PDEs with neural networks, where the most commonly used Dirichlet, Neumann, and Robin boundary conditions (BCs) are considered. Specifically, we first introduce the "extra fields" from the mixed finite element method to reformulate the PDEs so as to equivalently transform the three types of BCs into linear forms. Based on the reformulation, we derive the general solutions of the BCs analytically, which are employed to construct an ansatz that automatically satisfies the BCs. With such a framework, we can train the neural networks without adding extra loss terms and thus efficiently handle geometrically complex PDEs, alleviating the unbalanced competition between the loss terms corresponding to the BCs and PDEs. We theoretically demonstrate that the "extra fields" can stabilize the training process. Experimental results on real-world geometrically complex PDEs showcase the effectiveness of our method compared with state-of-the-art baselines.

## 1 Introduction

Many fundamental problems in science and engineering (e.g., [2, 19, 28]) are characterized by partial differential equations (PDEs) with the solution constrained by boundary conditions (BCs) that are derived from the physical system of the problem. Among all types of BCs, Dirichlet, Neumann, and Robin are the most commonly used [33]. Figure 1 gives an illustrative example on these three types of BCs. Furthermore, in practical problems, physical systems can be very geometrically complex (where the geometry of the definition domain is irregular or has complex structures, e.g., a lithium-ion battery [11], a heat sink [40], etc), leading to a large number of BCs. How to solve such PDEs has become a challenging problem shared by both the scientific and industrial communities.

The field of solving PDEs with neural networks has a history of more than 20 years [6, 1, 34, 8, 36]. Such methods are intrinsically mesh-free and therefore can handle high-dimensional as well as geometrically complex problems more efficiently compared with traditional mesh-based methods, like the finite element method (FEM). Physical-informed neural network (PINN) [25] is one of the most influential works, where the neural network is trained in the way of taking the residuals of both the PDEs and the BCs as multiple terms of the loss function. Although there are many wonderful improvements such as DPM [13], PINNs still face serious challenges as discussed in the paper [15]. Some theoretical works [37, 38] point out that there exists an unbalanced competition between the terms of PDEs and BCs, limiting the application of PINNs to geometrically complex problems. To address this issue, some researchers [3, 35, 18] have tried to embed BCs into the ansatz. Some of

---

*Corresponding author

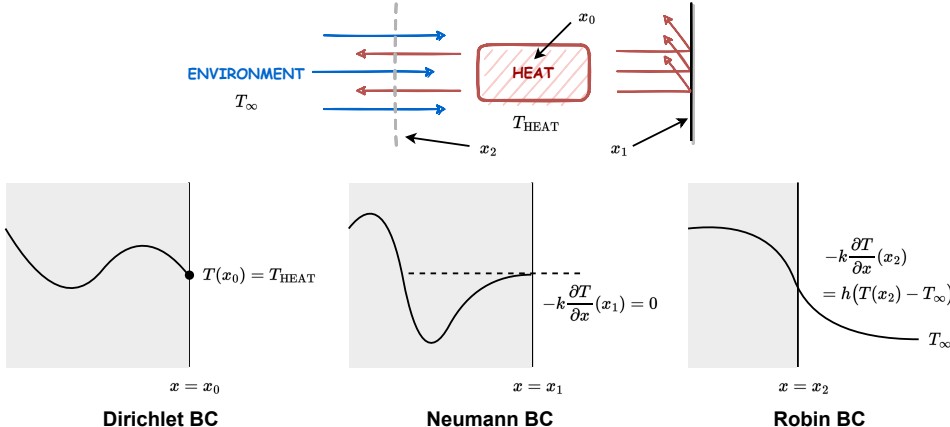

Figure 1: **An illustration on three types of BCs.** We give an example from heat transfer, where $T = T(x)$ is the temperature, $k$ is the thermal conductivity, and $h$ is the heat transfer coefficient. **(1)** The Dirichlet BC specifies the value of the solution at the boundary. Here we assume a constant temperature $T_{\text{HEAT}}$ at the heat source ($x = x_0$). **(2)** The Neumann BC specifies the value of the derivative at the boundary. We assume that the right wall ($x = x_2$) is adiabatic and force the derivative to be zero. **(3)** The Robin BC is a combination of the first two. And we use it to describe the heat convection between the heat source and the environment ($T_\infty$) at the surface ($x = x_2$).

them [16, 27] follow the pipeline of the Theory of Connections [22], while others [41, 12, 17] have considered solving the equivalent variational form of the PDEs. In this way, the neural networks can automatically satisfy the BCs and no longer require adding corresponding loss terms. Nevertheless, these methods are only applicable to specific BCs (e.g., Dirichlet BCs, homogeneous BCs, etc) or geometrically simple PDEs. The key challenge is that the equation forms of the Neumann and Robin BCs have no analytical solutions in general and are thus difficult to be embedded into the ansatz.

In this paper, we propose a unified hard-constraint framework for all the three most commonly used BCs (i.e., Dirichlet, Neumann and Robin BCs). With this framework (see Figure 2 for an illustration), we are able to construct an ansatz that automatically satisfies the three types of BCs. Therefore, we can train the model without the losses of these BCs, which alleviates the unbalanced competition and significantly improves the performance of solving geometrically complex PDEs. Specifically, we first introduce the *extra fields* from the mixed finite element method [20, 4]. This technique substitutes the gradient of a physical quantity with new variables, allowing the BCs to be reformulated as linear equations. Based on this reformulation, we derive a general continuous solution of the BCs with simple form, overcoming the challenge that the original BCs cannot be solved analytically. Using the general solutions obtained, we summarize a paradigm for constructing the hard-constraint ansatz under time-dependent, multi-boundary, and high-dimensional cases. Besides, in Section 4, we demonstrate that the technique of *extra fields* can improve the stability of the training process.

We empirically demonstrate the effectiveness of our method through three parts of experiments. First, we show the potency of our method in solving geometrically complex PDEs through two numerical experiments from real-world physical systems of a battery pack and an airfoil. And our framework achieves a supreme performance compared with advanced baselines, including the learning rate annealing methods [37], domain decomposition-based methods [10, 23], and existing hard-constraint methods [30, 31]. Second, we select a high-dimensional problem to demonstrate that our framework can be well applied to high-dimensional cases. Finally, we study the impact of the *extra fields* as well as some hyper-parameters and verify our theoretical results in Section 4.

To sum up, we make the following contributions:

- We introduce the *extra fields* to reformulate the PDEs, and theoretically demonstrate that our reformulation can effectively reduce the instability of the training process.
- We propose a unified hard-constraint framework for Dirichlet, Neumann, and Robin BCs, alleviating the unbalanced competition between losses in the physics-informed learning.
- Our method has superior performance over state-of-the-art baselines on solving geometrically complex PDEs, as validated by numerical experiments in real-world physical problems.

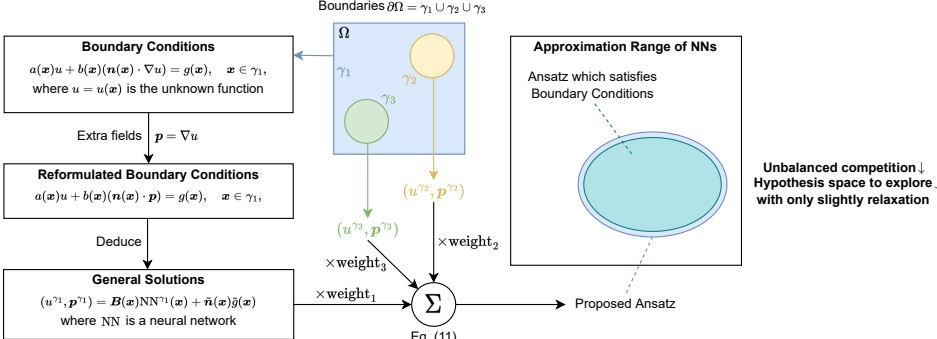

Figure 2: **A pipeline of proposed method.** In this paper, We consider PDEs with multiple boundary conditions of Dirichlet, Neumann, and Robin. We first introduce the *extra fields* to reformulate the BCs as linear equations whose general solutions are then deduced. Finally, we aggregate the general solutions for each boundary to obtain our final ansatz via Eq. (11). Since our ansatz automatically satisfies the BCs, the unbanlanced competition is alleviated and the hypothesis space is reduced.

## 2 Background

### 2.1 Physics-Informed Neural Networks (PINNs)

We consider the following Laplace's equation as a motivating example,

$$\Delta u(x_1, x_2) = 0, \qquad\qquad x_1 \in (0,1], x_2 \in [0,1], \tag{1a}$$
$$u(x_1, x_2) = g(x_2), \qquad\qquad x_1 = 0, x_2 \in [0,1], \tag{1b}$$

where Eq. (1a) gives the form of the PDE, and Eq. (1b) is a Dirichlet boundary condition (BC). A solution to the above problem is a solution to Eq. (1a) which also satisfies Eq. (1b).

Physics-informed neural networks (PINNs) [25] employ a neural network $\mathrm{NN}(x_1, x_2; \boldsymbol{\theta})$ to approximate the solution, i.e., $\hat{u}(x_1, x_2; \boldsymbol{\theta}) = \mathrm{NN}(x_1, x_2; \boldsymbol{\theta}) \approx u(x_1, x_2)$, where $\boldsymbol{\theta}$ denotes the trainable parameters of the network. And we learn the parameters $\boldsymbol{\theta}$ by minimizing the following loss function

$$\mathcal{L}(\boldsymbol{\theta}) = \mathcal{L}_{\mathcal{F}}(\boldsymbol{\theta}) + \mathcal{L}_{\mathcal{B}}(\boldsymbol{\theta}) \triangleq \frac{1}{N_f} \sum_{i=1}^{N_f} \left| \Delta \hat{u}(x_{f,1}^{(i)}, x_{f,2}^{(i)}; \boldsymbol{\theta}) \right|^2 + \frac{1}{N_b} \sum_{i=1}^{N_b} \left| \hat{u}(0, x_{b,2}^{(i)}; \boldsymbol{\theta}) - g(x_{b,2}^{(i)}) \right|^2, \tag{2}$$

where $\mathcal{L}_{\mathcal{F}}$ is the term restricting $\hat{u}$ to satisfy the PDE (Eq. (1a)) while $\mathcal{L}_{\mathcal{B}}$ is the BC (Eq. (1b)), $\{\boldsymbol{x}_f^{(i)} = (x_{f,1}^{(i)}, x_{f,2}^{(i)})\}_{i=1}^{N_f}$ is a set of $N_f$ collocation points sampled from $[0,1]^2$, and $\{\boldsymbol{x}_b^{(i)} = (0, x_{b,2}^{(i)})\}_{i=1}^{N_b}$ is a set of $N_b$ collocation points sampled from $x_1 = 0$.

PINNs have a wide range of applications, including heat [5], flow [21], and atmosphere [42]. However, PINNs are struggling with some issues on the performance [15]. Previous analysis [37, 38] has demonstrated that the convergence of $\mathcal{L}_{\mathcal{F}}$ can be significantly faster than that of $\mathcal{L}_{\mathcal{B}}$. As demonstrated in later experiments (see Section 5), this pathology may lead to nonphysical solutions which does not satisfy the BCs or initial conditions (ICs). Moreover, for geometrically complex PDEs where the number of BCs is large, this problem is exacerbated and can seriously affect accuracy, as supported by our experimental results in Table 1.

### 2.2 Hard-Constraint Methods

One potential approach to overcome this pathology is to embed the BCs into the ansatz in a way that any instance from the ansatz can automatically satisfy the BCs, as utilized by previous work [3, 24, 35, 39, 30, 18, 31]. We note that the loss terms corresponding to the BCs are no longer needed, and thus the above pathology is alleviated. These methods are called *hard-constraint methods*, and they share a similar formula of the ansatz as

$$\hat{u}(\boldsymbol{x}; \boldsymbol{\theta}) = u^{\partial \Omega}(\boldsymbol{x}) + l^{\partial \Omega}(\boldsymbol{x}) \mathrm{NN}(\boldsymbol{x}; \boldsymbol{\theta}), \tag{3}$$

where $\boldsymbol{x}$ is the coordinate, $\Omega$ is the domain of interest, $u^{\partial\Omega}(\boldsymbol{x})$ is the general solution at the boundary $\partial\Omega$, and $l^{\partial\Omega}(\boldsymbol{x})$ is an extended distance function which satisfies

$$l^{\partial\Omega}(\boldsymbol{x}) = \begin{cases} 0 & \text{if } \boldsymbol{x} \in \partial\Omega, \\ > 0 & \text{otherwise.} \end{cases} \tag{4}$$

In the case of Eq. (1) (where $\boldsymbol{x} = (x_1, x_2)$, $\Omega = [0,1]^2$), the general solution is exactly $g(x_2)$, and we can use the following ansatz (which automatically satisfies the BC in Eq. (1b))

$$\hat{u}(x_1, x_2; \boldsymbol{\theta}) = g(x_2) + x_1 \text{NN}(x_1, x_2; \boldsymbol{\theta}). \tag{5}$$

However, it is hard to directly extend this method to more general cases of Robin BCs (see Eq. (7)), since we cannot obtain the general solution $u^{\partial\Omega}(\boldsymbol{x})$ analytically. Existing attempts are either mesh-dependent [9, 43], which are time-consuming for high-dimensional and geometrically complex PDEs, or ad hoc methods for specific (geometrically simple) physical systems [26]. It is still lacking a unified hard-constraint framework for both geometrically complex PDEs and the most commonly used Dirichlet, Neumann, and Robin BCs.

## 3  Methodology

We first introduce the problem setup of geometrically complex PDEs considered in this paper and then reformulate the PDEs via the *extra fields*, followed by presenting our unified hard-constraint framework for embedding Dirichlet, Neumann and Robin BCs into the ansatz.

### 3.1  Problem Setup

We consider a physical system governed by the following PDEs defined on a geometrically complex domain $\Omega \subset \mathbb{R}^d$

$$\mathcal{F}[\boldsymbol{u}(\boldsymbol{x})] = \boldsymbol{0}, \qquad \boldsymbol{x} = (x_1, \ldots, x_d) \in \Omega, \tag{6}$$

where $\mathcal{F} = (\mathcal{F}_1, \ldots, \mathcal{F}_N)$ includes $N$ PDE operators which map $\boldsymbol{u}$ to a function of $\boldsymbol{x}$, $\boldsymbol{u}$ and its derivatives. Here, $\boldsymbol{u}(\boldsymbol{x}) = (u_1(\boldsymbol{x}), \ldots, u_n(\boldsymbol{x}))$ is the solution to the PDEs, which represents physical quantities of interest. For each $u_j, j = 1, \ldots, n$, we pose suitable boundary conditions (BCs) as

$$a_i(\boldsymbol{x})u_j + b_i(\boldsymbol{x})\big(\boldsymbol{n}(\boldsymbol{x}) \cdot \nabla u_j\big) = g_i(\boldsymbol{x}), \quad \boldsymbol{x} \in \gamma_i, \quad \forall i = 1, \ldots, m_j, \tag{7}$$

where $\gamma_1, \ldots, \gamma_{m_j}$ are subsets of the boundary $\partial\Omega$ whose closures are disjoint, $a_i(\boldsymbol{x})$ and $b_i(\boldsymbol{x})$ are two functions satisfying that $a_i^2(\boldsymbol{x}) + b_i^2(\boldsymbol{x}) \neq 0$ holds for $\forall \boldsymbol{x} \in \gamma_i$, and $\boldsymbol{n}(\boldsymbol{x}) = (n_1(\boldsymbol{x}), \ldots, n_d(\boldsymbol{x}))$ is the (outward facing) unit normal of corresponding $\gamma_i$ at $\boldsymbol{x}$. It is noted that Eq. (7) represents a Dirichlet BC if $a_i \equiv 1, b_i \equiv 0$, a Neumann BC if $a_i \equiv 0, b_i \equiv 1$, and a Robin BC otherwise.

For such geometrically complex PDEs, if we directly resort to PINNs (see Section 2.1), there would be a difficult multi-task learning with at least $(\sum_{j=1}^{n} m_j + N)$ terms in the loss function. As discussed in the previous analysis [37, 38], it will severely affect the convergence of the training due to the unbalanced competition between those loss terms. Hence, in this paper, we will discuss how to embed the BCs into the ansatz, where every instance automatically satisfies the BCs. However, it is infeasible to directly follow the pipeline of *hard-constraint methods* (see Eq. (3)) since Eq. (7) does not have an general solution of analytical form. Therefore, a new approach is needed to address this intractable problem.

### 3.2  Reformulating PDEs via Extra Fields

In this subsection, we present the general solutions of the BCs, which will be used to construct the hard-constraint ansatz subsequently. We first introduce the *extra fields* from the mixed finite element method [20, 4] to equivalently reformulate the PDEs. Let $\boldsymbol{p}_j(\boldsymbol{x}) = (p_{j1}(\boldsymbol{x}), \ldots, p_{jd}(\boldsymbol{x})) = \nabla u_j, j = 1, \ldots, n$. And we substitute them into Eq. (6) and Eq. (7) to obtain the equivalent PDEs,

$$\tilde{\mathcal{F}}[\boldsymbol{u}(\boldsymbol{x}), \boldsymbol{p}_1(\boldsymbol{x}), \ldots, \boldsymbol{p}_n(\boldsymbol{x})] = \boldsymbol{0}, \qquad \boldsymbol{x} \in \Omega, \tag{8a}$$

$$\boldsymbol{p}_j(\boldsymbol{x}) = \nabla u_j, \qquad \boldsymbol{x} \in \Omega \cup \partial\Omega, \qquad \forall j = 1, \ldots, n, \tag{8b}$$

where $(\boldsymbol{u}(\boldsymbol{x}), \boldsymbol{p}_1(\boldsymbol{x}), \ldots, \boldsymbol{p}_n(\boldsymbol{x}))$ is the solution of the new PDEs, $\tilde{\mathcal{F}} = (\tilde{\mathcal{F}}_1, \ldots, \tilde{\mathcal{F}}_N)$ are the PDE operators after the reformulation. And for $j = 1, \ldots, n$, we have the corresponding BCs

$$a_i(\boldsymbol{x})u_j + b_i(\boldsymbol{x})\big(\boldsymbol{n}(\boldsymbol{x}) \cdot \boldsymbol{p}_j(\boldsymbol{x})\big) = g_i(\boldsymbol{x}), \qquad \boldsymbol{x} \in \gamma_i, \qquad \forall i = 1, \ldots, m_j. \tag{9}$$

Now, we can see that Eq. (7) has been transformed into linear equations with respect to $(u_j, \boldsymbol{p}_j)$, which are much easier for us to derive general solutions. Hereinafter, we denote $(u_j, \boldsymbol{p}_j)$ by $\tilde{\boldsymbol{p}}_j$, and their general solutions at $\gamma_i$ (i.e., $(u_j^{\gamma_i}, \boldsymbol{p}_j^{\gamma_i})$) by $\tilde{\boldsymbol{p}}_j^{\gamma_i}$. Next we will discuss how to find $\tilde{\boldsymbol{p}}_j^{\gamma_i}$.

To obtain the general solution of Eq. (9), the first step is to find a basis $\boldsymbol{B}(\boldsymbol{x})$ of the null space (whose dimension is $d$). However, we must emphasize that $\boldsymbol{B}(\boldsymbol{x})$ should be carefully chosen. Since Eq. (9) is parameterized by $\boldsymbol{x}$, for any $\boldsymbol{x} \in \gamma_i$, $\boldsymbol{B}(\boldsymbol{x})$ should always be a basis of the null space, that is, its columns cannot degenerate into linearly dependent vectors (otherwise it will not be able to represent all possible solutions). An example of an inadmissible $\boldsymbol{B}(\boldsymbol{x})$ is given in Appendix A.1.

Generally, for any dimension $d \in \mathbb{R}$, we believe it is non-trivial to find a simple expression for the basis. Instead, we prefer to find $(d + 1)$ vectors in the null space, $d$ of which are linearly independent (that way, $\boldsymbol{B}(\boldsymbol{x}) \in \mathbb{R}^{(d+1) \times (d+1)}$). We now directly present our construction of the general solution while leaving a detailed derivation in Appendix A.3.

$$\tilde{\boldsymbol{p}}_j^{\gamma_i}(\boldsymbol{x}; \boldsymbol{\theta}_j^{\gamma_i}) = \boldsymbol{B}(\boldsymbol{x})\mathrm{NN}_j^{\gamma_i}(\boldsymbol{x}; \boldsymbol{\theta}_j^{\gamma_i}) + \tilde{\boldsymbol{n}}(\boldsymbol{x})\tilde{g}_i(\boldsymbol{x}), \tag{10}$$

where $\tilde{\boldsymbol{n}} = (a_i, b_i\boldsymbol{n})\big/\sqrt{a_i^2 + b_i^2}$, $\tilde{g}_i = g_i\big/\sqrt{a_i^2 + b_i^2}$, $\mathrm{NN}_j^{\gamma_i} : \mathbb{R}^d \to \mathbb{R}^{d+1}$ is a neural network with trainable parameters $\boldsymbol{\theta}_j^{\gamma_i}$, and $\boldsymbol{B}(\boldsymbol{x}) = \boldsymbol{I}_{d+1} - \tilde{\boldsymbol{n}}(\boldsymbol{x})\tilde{\boldsymbol{n}}(\boldsymbol{x})^\top$. Incidentally, in the case of $d = 1$ or $d = 2$, we can find a simpler expression for $\boldsymbol{B}$ (see Appendix A.2). Besides, we note that all the neural networks are implemented as MLPs in this paper, unless otherwise stated.

### 3.3 A Unified Hard-Constraint Framework

With the parameterization of a neural network, Eq. (10) can represent any function in $\gamma_i$, as long as the function satisfies the BC (see Eq. (9)). Since our problem domain contains multiple boundaries, we need to combine the general solutions corresponding to each boundary $\gamma_i$ to achieve an overall approximation. Hence, we construct our ansatz as follows

$$(\hat{u}_j, \hat{\boldsymbol{p}}_j) = l^{\partial\Omega}(\boldsymbol{x})\mathrm{NN}_{\mathrm{main}}(\boldsymbol{x}; \boldsymbol{\theta}_{\mathrm{main}}) + \sum_{i=1}^{m_j} \exp\big[-\alpha_i l^{\gamma_i}(\boldsymbol{x})\big]\tilde{\boldsymbol{p}}_j^{\gamma_i}(\boldsymbol{x}; \boldsymbol{\theta}_j^{\gamma_i}), \quad \forall j = 1, \ldots, n \tag{11}$$

where $\mathrm{NN}_{\mathrm{main}} : \mathbb{R}^d \to \mathbb{R}^{d+1}$ is the main neural network with trainable parameters $\boldsymbol{\theta}_{\mathrm{main}}$, $l^{\partial\Omega}, l^{\gamma_i}, i = 1, \ldots, m_j$ are continuous extended distance functions (see Eq. (4)), and $\alpha_i$ ($i = 1, \ldots, m_j$) are determined by

$$\alpha_i = \frac{\beta_s}{\min_{\boldsymbol{x} \in \partial\Omega \backslash \gamma_i} l^{\gamma_i}(\boldsymbol{x})}, \tag{12}$$

where $\beta_s \in \mathbb{R}$ is a hyper-parameter of the "hardness" in the spatial domain. In Eq. (11), we utilize extended distance functions to "divide" the problem domain into several parts, where $\tilde{\boldsymbol{p}}_j^{\gamma_i}$ ($\mathrm{NN}_j^{\gamma_i}$ is its learnable part) is responsible for the approximation on the boundaries while $\mathrm{NN}_{\mathrm{main}}$ is responsible for internal. Furthermore, Eq. (12) ensures that the weight of $\tilde{\boldsymbol{p}}_j^{\gamma_i}$ decays to $e^{-\beta_s}$ at the nearest neighbor of $\gamma_i$, so that $\tilde{\boldsymbol{p}}_j^{\gamma_i}$ does not interfere with the approximation on other boundaries. We provide a theoretical guarantee of the satisfaction of the BCs along with the approximation ability for Eq. (11) in Appendix A.4. Besides, if $a_i, b_i, \boldsymbol{n}$ or $g_i$ are only defined at $\gamma_i$, we can extend their definition to $\Omega \cup \partial\Omega$ using interpolation or approximation via neural networks (see Appendix A.5). And it is easy to extend this framework to the spatial-temporal domain as discussed in Appendix A.6.

Finally, we can train our model with the following loss function

$$\mathcal{L} = \frac{1}{N_f} \sum_{k=1}^{N_f} \sum_{j=1}^{N} \big|\tilde{\mathcal{F}}_j[\hat{\boldsymbol{u}}(\boldsymbol{x}^{(k)}), \hat{\boldsymbol{p}}_1(\boldsymbol{x}^{(k)}), \ldots, \hat{\boldsymbol{p}}_n(\boldsymbol{x}^{(k)})]\big|^2$$

$$+ \frac{1}{N_f} \sum_{k=1}^{N_f} \sum_{j=1}^{n} \big\|\hat{\boldsymbol{p}}_j(\boldsymbol{x}^{(k)}) - \nabla\hat{u}_j(\boldsymbol{x}^{(k)})\big\|_2^2, \tag{13}$$

where $\hat{\boldsymbol{u}} = (\hat{u}_1, \ldots, \hat{u}_n)$, $\{\hat{u}_j, \hat{\boldsymbol{p}}_j\}_{j=1}^{n}$ is defined in Eq. (11) and $\{\boldsymbol{x}^{(k)}\}_{k=1}^{N_f}$ is a set of collocation points sampled in $\Omega$. For neatness, we omit the trainable parameters of neural networks here. We note that Eq. (13) measures the discrepancy of both the PDEs (i.e., $\tilde{\mathcal{F}}_1, \ldots, \tilde{\mathcal{F}}_N$) and the equilibrium equations introduced by the *extra fields* (i.e., Eq. (8b)) at $N_f$ collocation points.

According to Eq. (13), we have now successfully embedded BCs into the ansatz, and no longer need to take the residuals of BCs as extra terms in the loss function. That is, our ansatz strictly conforms to BCs throughout the training process, greatly reducing the possibility of generating nonphysical solutions. Nevertheless, this comes at the cost of introducing $(nd)$ additional equilibrium equations (see Eq. (8b)). But in many physical systems, especially those with complex geometries, the number of BCs (cnt(BCs)) is far larger than $(nd)$ (e.g., $n = 3$, $d = 2$, cnt(BCs) = 1260 for a classical physical system, a heat exchanger [7]). So we may actually reduce the number of loss terms by an order of magnitude or two ($\Delta$cnt(losses) = $nd - $cnt(BCs) $\ll 0$), alleviating the unbalanced competition between loss terms.

## 4 Theoretical Analysis

In Section 3.2, we introduce the *extra fields* and reformulate the PDEs (from Eq. (6) and Eq. (7) to Eq. (8) and Eq. (9)). To further analyze the impact of this reformulation, we consider the following abstraction of 1D PDEs (Eq. (14a)) and the one after the reformulation (Eq. (14b))

$$L\Big[u, \frac{\mathrm{d}u}{\mathrm{d}x}, \frac{\mathrm{d}^2 u}{\mathrm{d}x^2}, \cdots, \frac{\mathrm{d}^n u}{\mathrm{d}x^n}\Big] = f(x), \qquad x \in \Omega, \tag{14a}$$

$$L\big[u, p_1, p_2, \cdots, p_{n-1}\big] = f(x), \qquad x \in \Omega, \tag{14b}$$

where $L$ is an operator, $f$ is a source function, and $p_{i+1} = \mathrm{d}p_i/\mathrm{d}x, p_1 = \mathrm{d}u/\mathrm{d}x$ are the introduced extra fields. From the above equations, we can find that the orders of derivatives in the PDEs are reduced. Intuitively, as the PDEs are included in the loss function (see Eq. (2)), lower derivatives result in less accumulation of back-propagation, and thus stabilize the training process.

To formally explain this mechanism, without loss of generality, we consider a simple case (with proper Dirichlet BCs) where $L[\cdots] = \Delta u$, $f(x) = -a^2 \sin ax$. And our ansatz is a single-layer neural network of width $K$, i.e., $\hat{u} = \boldsymbol{c}^\top \sigma(\boldsymbol{w}x + \boldsymbol{b})$, $\hat{p} = \boldsymbol{c}_p^\top \sigma(\boldsymbol{w}x + \boldsymbol{b})$, where $\boldsymbol{c}, \boldsymbol{w}, \boldsymbol{b}, \boldsymbol{c}_p \in \mathbb{R}^K$, $\sigma$ is an element-wise activation function (for simplicity, we take $\sigma$ as $\tanh$). More details on the PDEs are given in Appendix A.7.1. Next, we focus on the gradients of the loss terms of the PDEs $\mathcal{F}$, since that of the BC stays the same during the reformulation. We state the following theorem.

**Theorem 4.1** (Bounds for the Gradients of the PDE Loss Terms). *Let $\boldsymbol{\theta} = (\boldsymbol{c}, \boldsymbol{w}, \boldsymbol{b})$, $\tilde{\boldsymbol{\theta}} = (\boldsymbol{c}, \boldsymbol{w}, \boldsymbol{b}, \boldsymbol{c}_p)$, and $\mathcal{L}_\mathcal{F}$ as well as $\tilde{\mathcal{L}}_\mathcal{F}$ be the loss terms corresponding to the original and transformed PDEs, respectively. We have the following bounds*

$$\Big|\big(\nabla_{\boldsymbol{\theta}} \mathcal{L}_\mathcal{F}\big)^\top\Big| = \mathcal{O}\big(|\boldsymbol{c}|^\top \boldsymbol{w}^2 + a^2\big) \cdot \big(\boldsymbol{w}^2, |\boldsymbol{c}| \circ |\boldsymbol{w}| \circ (|\boldsymbol{w}| + \boldsymbol{1}), |\boldsymbol{c}| \circ \boldsymbol{w}^2\big), \tag{15a}$$

$$\Big|\big(\nabla_{\tilde{\boldsymbol{\theta}}} \tilde{\mathcal{L}}_\mathcal{F}\big)^\top\Big| = \mathcal{O}\big(\|\boldsymbol{c}_p\|_1 + \max(|\boldsymbol{c}|, |\boldsymbol{c}_p|)^\top |\boldsymbol{w}| + a^2\big) \cdot \big(|\boldsymbol{w}|, \max(|\boldsymbol{c}|, |\boldsymbol{c}_p|) \circ \max(|\boldsymbol{w}|, \boldsymbol{1}),$$

$$\max(|\boldsymbol{c}|, |\boldsymbol{c}_p|) \circ \max(|\boldsymbol{w}|, \boldsymbol{1}), \max(|\boldsymbol{w}|, \boldsymbol{1})\big), \tag{15b}$$

*where $\circ$ is the element-wise multiplication, $\boldsymbol{1}$ is an all-ones vector, $\max(\cdot)$ is the element-wise maximum of vectors, and operations on vectors ($|\cdot|$, $(\cdot)^2$, etc) are element-wise operations.*

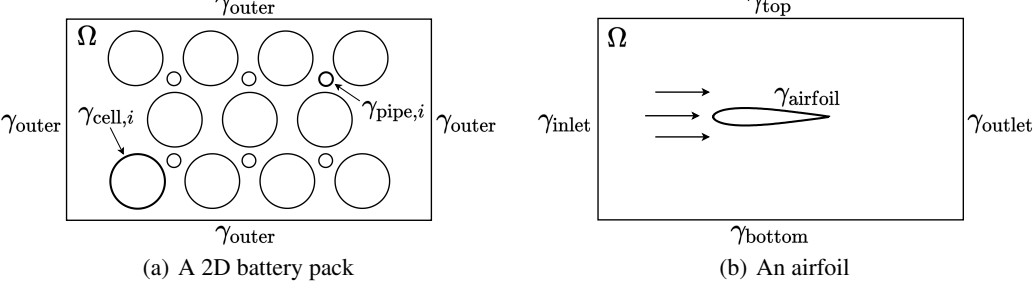

(a) A 2D battery pack           (b) An airfoil

Figure 3: An illustration of the geometries of the problems in Section 5.2 and Section 5.3.

The proof is deferred to Appendix A.7.2. We can find that $(\nabla_{\boldsymbol{\theta}}\mathcal{L}_{\mathcal{F}})^{\top}$ has a high-order relationship with the trainable parameters $\boldsymbol{\theta}$. This often makes training unstable, as the gradients $(\nabla_{\boldsymbol{\theta}}\mathcal{L}_{\mathcal{F}})^{\top}$ may rapidly explode or vanish when $\boldsymbol{\theta}$ are large or small. After the reformulation, $(\nabla_{\tilde{\boldsymbol{\theta}}}\tilde{\mathcal{L}}_{\mathcal{F}})^{\top}$ is associated with the parameters in a lower order, and some terms are controlled by 1, alleviating this issue of the instability. We note that the improvement in stability comes from lower derivatives in the PDEs, and this mechanism does not depend on the specific form of the PDEs. In Section 5.5.1, we empirically show that this mechanism is general, even for nonlinear PDEs.

# 5 Experiments

In this section, we will numerically showcase the effectiveness of the proposed framework on geometrically complex PDEs, which is also the main focus of this paper. To this end, we test our framework on two real-world problems (see Section 5.2 and Section 5.3) from a 2D battery pack and an airfoil, respectively. Besides, to validate that our framework is applicable to high-dimensional cases, an additional high-dimension problem is considered in Section 5.4, followed by an ablation study in Section 5.5. We refer to Appendix A.8 for experimental details.

## 5.1 Experiment Setup

**Evaluation**   In our experiments, we utilize mean absolute error (MAE) and mean absolute percentage error (MAPE) to measure the discrepancy between the prediction and the ground truth. In Section 5.3, we replace MAPE with weighted mean absolute percentage error (WMAPE) to avoid the "division by zero", since a considerable part of the ground truth values are very close to zero,

$$\text{WMAPE} = \frac{\sum_{i=1}^{n}|\hat{y}_i - y_i|}{\sum_{i=1}^{n}|y_i|}, \tag{16}$$

where $n$ is the number of testing samples, $\hat{y}_i$ is the prediction and $y_i$ is the ground truth.

**Baselines**   We consider the following baselines in subsequent experiments.

- **PINN**: original implementation of the PINN [25].
- **PINN-LA & PINN-LA-2**: PINN with learning rate annealing algorithm to address the unbalanced competition [37] and a variant we modified for numerical stability.
- **xPINN & FBPINN**: two representative works for solving geometrically complex PDEs via non-overlapping [10] and overlapping [23] domain decomposition.
- **PFNN & PFNN-2**: a representative hard-constraint method based on the variational formulation of PDEs [30] and its latest variant incorporating domain decomposition [31].

## 5.2 Simulation of a 2D battery pack (Heat Equation)

To emphasize the capability of the proposed hard-constraint framework (HC) to handle geometrically complex cases, we first consider modeling thermal dynamics of a 2D battery pack [32], which is

Table 1: Experimental results of the simulation of a 2D battery pack

|  | MAE of $T$ | | | | MAPE of $T$ | | | |
|---|---|---|---|---|---|---|---|---|
|  | $t=0$ | $t=0.5$ | $t=1$ | average | $t=0$ | $t=0.5$ | $t=1$ | average |
| PINN | 0.1283 | 0.0457 | 0.0287 | 0.0539 | 128.21% | 11.65% | 4.47% | 24.82% |
| PINN-LA | 0.0918 | 0.0652 | 0.0621 | 0.0661 | 91.72% | 19.13% | 11.96% | 27.06% |
| PINN-LA-2 | 0.1062 | 0.0321 | **0.0211** | 0.0402 | 106.05% | 8.94% | 4.09% | 19.76% |
| FBPINN | 0.0704 | 0.0293 | 0.0249 | 0.0343 | 70.33% | 8.13% | 5.87% | 14.74% |
| xPINN | 0.2230 | 0.1295 | 0.1515 | 0.1454 | 222.83% | 30.28% | 20.25% | 54.70% |
| PFNN | **0.0000** | 0.3036 | 0.4308 | 0.2758 | **0.02%** | 79.64% | 84.60% | 68.29% |
| PFNN-2 | **0.0000** | 0.3462 | 0.5474 | 0.3215 | **0.02%** | 66.06% | 90.21% | 59.62% |
| HC | **0.0000** | **0.0246** | 0.0225 | **0.0221** | **0.02%** | **5.38%** | **3.77%** | **5.10%** |

Table 2: Experimental results of the simulation of an airfoil

| | MAE | | | WMAPE | | |
|---|---|---|---|---|---|---|
| | $u_1$ | $u_2$ | $p$ | $u_1$ | $u_2$ | $p$ |
| PINN | 0.4682 | 0.0697 | 0.3883 | 0.5924 | 1.1979 | 0.3539 |
| PINN-LA | 0.4018 | 0.0595 | 0.2652 | 0.5084 | 1.0225 | 0.2418 |
| PINN-LA-2 | 0.5047 | 0.0659 | 0.2765 | 0.6385 | 1.1325 | 0.2521 |
| FBPINN | 0.4058 | 0.0563 | 0.2665 | 0.5134 | 0.9676 | 0.2429 |
| xPINN | 0.7188 | 0.0583 | 1.1708 | 0.9095 | 1.0029 | 1.0672 |
| HC | **0.2689** | **0.0435** | **0.2032** | **0.3402** | **0.7474** | **0.1852** |

governed by the heat equation (see Appendix A.8.1), where $x \in \Omega, t \in [0, 1]$ are the spatial and temporal coordinates, respectively, $T(x, t)$ is the temperature over time. The geometry of the problem $\Omega$ is shown in Figure 3(a), where $\gamma_{\text{cell},i}$ is the cell and $\gamma_{\text{pipe},i}$ is the cooling pipe.

The baselines are trained with $N_f = 8192$ collocation points, $N_b = 512$ boundary points and $N_i = 512$ initial points (an additional $N_s = 512$ points on the interfaces between subdomains are required for the xPINN), while the HC is trained with $N_f = 8192$ collocation points. All the models are trained for 5000 Adam [14] iterations (with a learning rate scheduler) followed by a L-BFGS optimization until convergence and tested with $N_t = 146,487$ points evaluated by the FEM. The testing results are given in Table 1 (where "average" means averaging over $t \in [0, 1]$). We find that the accuracy of the HC is significantly higher than baselines almost all the time, especially at $t = 0$. This is because the ansatz of the HC always satisfies the BCs and ICs during the training process, prevents the approximations from violating physical constraints at the boundaries. Besides, we also provide an empirical analysis of convergence in Appendix A.9 and results of 5-run parallel experiments (where problems in all three cases are revisited) in Appendix A.10.

## 5.3 Simulation of an Airfoil (Navier-Stokes Equations)

In the next experiment, we consider the challenging benchmark PDEs in computational fluid dynamics, the 2D stationary incompressible Navier-Stokes equations, in the context of simulating the airflow around a real-world airfoil (`w1015.dat`) from the UIUC airfoil coordinates database (an open airfoil database) [29]. Specifically, the airfoil is represented by 240 anchor points and the governing equation can be found in Appendix A.8.2, where $u(x) = (u_1(x), u_2(x))$, $p(x)$ are the velocity, and the pressure of the fluid, respectively, and the geometry of the problem is shown in Figure 3(b).

The baselines (PFNN and PFNN-2 are not applicable to Navier-Stokes Equations) are trained with $N_f = 10,000$ collocation points, $N_b = 2048$ boundary points ($N_s = 2048$ points for the xPINN), while the HC is trained with $N_f = 10,000$ collocation points. All the models are trained for 5000 Adam iterations followed by a L-BFGS optimization until convergence and tested with $N_t = 13,651$ points evaluated by the FEM. According to the results in Table 2, the HC has more obvious advantages, compared with the previous experiment. This is because complex nonlinear PDEs are more sensitive to the BCs, and the failure at the BCs can cause the approximations to deviate significantly from the true solutions. In addition, we find that the WMAPE on $u_2$ is relatively large for all methods, which is because the true values of $u_2$ are close to zero, amplifying the effect of the noise.

Table 3: Experimental results of the high-dimensional heat equation

| | MAE of $u$ | | | | MAPE of $u$ | | | |
|---|---|---|---|---|---|---|---|---|
| | $t = 0$ | $t = 0.5$ | $t = 1$ | average | $t = 0$ | $t = 0.5$ | $t = 1$ | average |
| PINN | 0.0219 | 0.0428 | 0.1687 | 0.0582 | 1.43% | 1.70% | 4.04% | 1.99% |
| PINN-LA | 0.0085 | 0.0149 | 0.0727 | 0.0235 | 0.55% | 0.59% | 1.73% | 0.78% |
| PINN-LA-2 | 0.0122 | 0.0274 | 0.1495 | 0.0466 | 0.79% | 1.08% | 3.57% | 1.49% |
| PFNN | **0.0000** | 0.1253 | 0.3367 | 0.1425 | **0.00%** | 5.02% | 8.19% | 4.64% |
| HC | **0.0000** | **0.0029** | **0.0043** | **0.0026** | **0.00%** | **0.12%** | **0.11%** | **0.10%** |

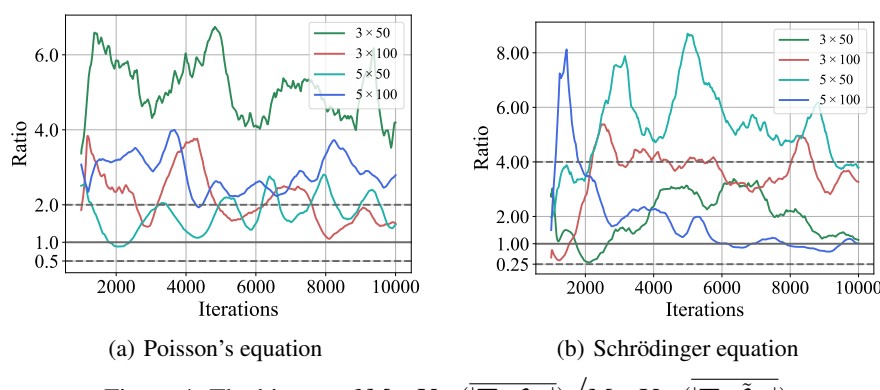

(a) Poisson's equation        (b) Schrödinger equation

Figure 4: The history of $\mathrm{MovVar}(\overline{|\nabla_{\boldsymbol{\theta}}\mathcal{L}_{\mathcal{F}}|})\Big/\mathrm{MovVar}(\overline{|\nabla_{\tilde{\boldsymbol{\theta}}}\tilde{\mathcal{L}}_{\mathcal{F}}|})$.

## 5.4 High-Dimensional Heat Equation

To demonstrate that our framework can generalize to high-dimensional cases, we consider a problem of time-dependent high-dimensional PDEs in $\Omega \times [0,1]$ (see Appendix A.8.3), where $\Omega$ is a unit ball in $\mathbb{R}^{10}$, and $u$ is the quantity of interest.

Here, we do not consider the baselines of the xPINN, FBPINN, and PFNN-2, because the method of domain decomposition is not suitable for high-dimensional cases. And the other baselines are trained with $N_f = 1000$ collocation points, $N_b = 100$ boundary points and $N_i = 100$ initial points, while the HC is trained with $N_f = 1000$ collocation points. All the models are trained for 5000 Adam iterations followed by a L-BFGS optimization until convergence and tested with $N_t = 10,000$ points evaluated by the analytical solution to the PDEs. We refer to Table 3 for the testing results. It can be seen that the HC achieves impressive performance in high-dimensional cases. This illustrates the effectiveness of Eq. (10) and the fact that our model can be directly applied to high-dimensional PDEs, not just geometrically complex PDEs.

## 5.5 Ablation Study

In this subsection, we are interested in the impact of the "extra fields" and the hyper-parameters of "hardness" (i.e., $\beta_s$ and $\beta_t$, see Section 3.3).

### 5.5.1 Extra fields

We test the PINNs (with and without the "extra fields") on the Poisson's equation and the nonlinear Schrödinger equation (see Appendix A.8.4). We vary the network architecture and report the ratio of the the moving variance (MovVar) of $\overline{|\nabla_{\boldsymbol{\theta}}\mathcal{L}_{\mathcal{F}}|}$ to that of $\overline{|\nabla_{\tilde{\boldsymbol{\theta}}}\tilde{\mathcal{L}}_{\mathcal{F}}|}$ at each iteration (see Figure 4). In both linear and nonlinear PDEs, we find that models with the "extra fields" ($\overline{|\nabla_{\tilde{\boldsymbol{\theta}}}\tilde{\mathcal{L}}_{\mathcal{F}}|}$) achieve smaller gradient oscillations during training, which is consistent with our theoretical results in Section 4. This is because the "extra fields" reduces the orders of the derivatives, which in turn reduces the accumulation of backpropagation, avoiding vanishing or exploding gradients.

### 5.5.2 Hyper-parameters of Hardness

We repeated the experiment in Section 5.2 with different combinations of the values of $\beta_s$ and $\beta_t$, and all other settings stay the same. Table 4 gives the testing results (where the MAE and MAPE

Table 4: The MAE / MAPE of $T$ on different $\beta_s$ and $\beta_t$

|  | $\beta_t = 1$ | $\beta_t = 2$ | $\beta_t = 5$ | $\beta_t = 10$ |
|---|---|---|---|---|
| $\beta_s = 1$ | 0.3492 / 48.21% | 0.3539 / 48.56% | 0.3226 / 44.64% | 0.2889 / 40.69% |
| $\beta_s = 2$ | 0.2800 / 40.30% | 0.1670 / 26.16% | 0.2140 / 31.72% | 0.1619 / 25.20% |
| $\beta_s = 5$ | 0.1878 / 28.68% | 0.1195 / 19.68% | 0.0542 / 10.35% | **0.0221 / 5.10%** |
| $\beta_s = 10$ | 0.1896 / 29.15% | 0.1104 / 18.70% | 0.0517 / 10.56% | 0.0329 / 8.15% |

are the average results over $t \in [0, 1]$). In general, the accuracy increases as the values of $\beta_s$ and $\beta_t$ become larger. The reason is that "harder" $\beta_s$ and $\beta_t$ cause the BCs to be better satisfied, which in turn makes the approximations more compliant with physical constraints. However, too large $\beta_s$ and $\beta_t$ (they are both on the exponential, see Eq. (11)) can lead to performance degradation, which may be related to numerical stability.

## 6  Conclusion

In this paper, we develop a unified hard-constraint framework for solving geometrically complex PDEs. With the help of "extra fields", we reformulate the PDEs and find the general solutions of the Dirichlet, Neumann, and Robin BCs. Based on this derivation, we propose a general formula of the hard-constraint ansatz which is applicable to time-dependent, multi-boundary, and high-dimensional cases. Besides, our theoretical analysis demonstrates that the reformulation is helpful for training stability. Extensive experiments show that our method can achieve state-of-the-art performance in real-world geometrically complex as well as high-dimension PDEs, and our theoretical results are universal to general PDEs. One of the limitations is that here we only consider the three most commonly used BCs and future works may include extending the framework to more general BCs.

## Acknowledgements

This work was supported by the National Key Research and Development Program of China (2017YFA0700904, 2020AAA0106000, 2020AAA0104304, 2020AAA0106302), NSFC Projects (Nos. 62061136001, 62076145, 62076147, U19B2034, U1811461, U19A2081, 61972224), Beijing NSF Project (No. JQ19016), BNRist (BNR2022RC01006), Tsinghua Institute for Guo Qiang, and the High Performance Computing Center, Tsinghua University. J.Z is also supported by the XPlorer Prize.

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
