# OpenReview forum: "A Unified Hard-Constraint Framework for Solving Geometrically Complex PDEs"
_NeurIPS.cc/2022/Conference — NeurIPS 2022 Accept_

### Official Review · Reviewer_HVPE · 2022-07-05

**Rating:** 3
**Confidence:** 4
**Soundness:** 3 good
**Presentation:** 2 fair
**Contribution:** 2 fair

**Summary:**

The submission introduces a new technique to enforce Dirichlet, Neumann, and Robin boundary constraints into the PINNs model, based on the extra fields technique from the mixed finite element method. Extra fields decomposes the solution into a linear combination of a boundary term, which is given to us as a constraint, and an interior term, learned by a neural network. On the boundary, the interpolation is designed so that only the boundary term is accounted for in the solution. As such, this is a handcrafted hard constraint method. The authors test on 3 PDEs: the heat equation in a battery pack, the Navier-Stokes equation over an airfoil, and a 10 dimensional heat equation, comparing against a variety of PINNs-based models. They claim that their method is stabler to train.

**Questions:**

“Geometrically complex” -  this term is used a lot and yet not defined anywhere. It would be useful to have a definition or at least a description of what this could mean.

23 - It is stated that neural networks have only recently been applied to the solution of PDEs, but the field is in fact older, dating back more than 20 years.

Eqn 1a:  is the domain of x2 correct? I think it may be x2 \in (0,1].

107 - what is meant by “PDE operator”? I think I understand, but a clear definition would be useful. Do you not just mean a differential operator?

133: it would be useful to provide an example where you show how the nullspace of eqn. 9 is computed.

Eqn 10 - what architecture is used for NN_i? This is missing from the paper and highly influences the results that can be achieved.

Sec 3.3 - Would it be possible to achieve the decomposition of the solution	 into a hard-coded boundary term and a learned interior term without the introduction of the extra fields per se? In eqn 13 you have to add a loss term encouraging p to match \nabla u. Instead, you could have just set p to be \nabla u itself and trained this directly. Or is it too difficult to achieve this given that p itself has to live in the nullspace of eqn 9?

182 -  I think you mean width, not length

Eqn 16:  where are the weights?

Experiments: it would be useful for the boundary conditions to be explicitly shown in this section, along with the architectures used, how the nullspace basis is solved for and the choice of extended distance function l.

Tables: were the results computed over multiple runs? It would have been useful/more significant to have confidence internals/standard deviations shown, so that the significance of each result could be ascertained.

Typos:
13 - effeteness
63 - alliviating
138 - non-trival
203 - addition -> additional

**Limitations:**

Limitations of this work are not addressed. It would be useful to write a paragraph explaining the shortcomings of having to solve for the nullspace of the boundary condition equations and other limitations of using PINNs as a baseline.

**Strengths And Weaknesses:**

+ The mathematics and derivations appear to be sound (although a little hard to follow) and it appears from the experimental section that this technique is better than competing methods.

+ The method is grounded in a standard technique from the field of mixed finite element solvers, and so it inherits a lot of the benefits of that method.

+ The submission steps the reader through a lot of needed material and builds up intuition.

- It is hard to gauge the significance of the method, since there are no confidence intervals in the experiment section

- I could not reimplement this submission. The main paper lacks many essential details.

-  In many places, the writing lacks clarity. Terms like geometrically complex or PDE operators are used,  without clear definitions. This should, however, be easy to clear up.

---

> ### Author Response · Authors · 2022-08-02
> **Thanks for your valuable comments (Part II)**
>
> ## Q5: Several writing issues.
>
> It is very nice of you to offer detailed writing suggestions. And we have thought over all of them carefully and have made the following modifications in the latest version of the paper after rebuttal revision. Thank you.
>
> - **The description of "geometrically complex".** In this paper, the term "geometrically complex" mainly refers to the irregular and complex geometry of the definition domain of the physical system/PDEs (e.g., many-sided polygons, porous structures, etc). Examples of geometrically complex physical systems include a lithium-ion battery [1] and a heat sink [2]. Mathematically, the "geometrically complex" is mainly manifested in that the boundary is irregular or the number of the BCs is large. According to your advice, we have added a proper description in the main text (see **Line 20**).
> - **The history of applying neural networks in solving PDEs.** We have revised the introduction. See Line **23**.
> - **The domain of Eq. (1a).** I guess that you mean the domain of Eq. (1a) and (1b) should not intersect. We have revised this formulation. See **Eq. (1)**.
> - **The description of "PDE operator".** We adopt the term "PDE operator" from the paper [3]. We agree that it will be useful to add a clear description. See **Line 107** for our modification.
> - **The derivation of the nullspace of Eq. (9).** We totally agree with you. In fact, we have presented an admissible general solution to Eq. (9) in Eq. (10). However, it may be a little hard to follow. Hence, we have provided a detailed derivation of Eq. (10) in **Appendix A.3**.
> - **The architecture of $\mathrm{NN}_i$ in Eq. (10).** Actually, the architecture of every neural network mentioned in the paper is MLP, which is given in the experimental details (see **Appendix A.8**). However, we agree with you that it is necessary to introduce the architecture of the network in the main text. So we have added this on **Line 144** in our latest version.
> - **The word "length" in Line 182.** Yes, you are right. Thank you for your reminder. We have corrected this error. See **Line 186**.
> - **The weights in Eq. (16).** We refer to Wikipedia (https://en.wikipedia.org/wiki/WMAPE) for the definition of weighted mean absolute percentage error (WMAPE). Here, the word "weighted" indicates that errors are weighted by values of actuals.
> - **The BCs along with architectures used in experiments.** Due to the page limit, we have left further experimental details in the **Appendix A.8**. Besides, we have added richer experimental details for readers' reference. Please refer to **Appendix A.8** in our latest version.
> - **Several typos.** Thanks for your careful review. We have corrected all the typos. See **Line 13, 63, 137, and 207**.
>
> ## References
>
> [1] Jeon, D. H., & Baek, S. M. (2011). Thermal modeling of cylindrical lithium ion battery during discharge cycle. Energy Conversion and Management, 52(8-9), 2973-2981.
>
> [2] Wu, T., Ozpineci, B., Chinthavali, M., Wang, Z., Debnath, S., & Campbell, S. (2017, June). Design and optimization of 3D printed air-cooled heat sinks based on genetic algorithms. In 2017 IEEE Transportation Electrification Conference and Expo (ITEC) (pp. 650-655). IEEE.
>
> [3] Wang, S., Teng, Y., & Perdikaris, P. (2021). Understanding and mitigating gradient flow pathologies in physics-informed neural networks. SIAM Journal on Scientific Computing, 43(5), A3055-A3081.

---

> ### Author Response · Authors · 2022-08-02
> **Thanks for your valuable comments (Part I)**
>
> ## Q1: Need for confidence intervals in the experimental results.
>
> Thanks for your advice. We totally agree that it is very important to have confidence intervals since many machine learning-based methods inherently have randomness. Hence, we have performed parallel experiments (in 5 runs) over the three cases (in Section 5.2, 5.3, and 5.4). The testing results are provided in **Appendix A.10** in our latest version (we put our appendix along with source codes in the supplementary material). From the results, we find that our method, HC still outperforms other baselines and has the least variation.
>
> ## Q2: Necessity of adding $\boldsymbol{p}=\nabla \boldsymbol{u}$ into the loss function.
>
> We consider the abstraction of the three types of BCs in Eq. (7). The form of this equation is a first-order linear PDE, where the parameters $a_i,b_i,\boldsymbol{n},g_i$ can be arbitrary functions and the general solution is mathematically difficult to construct. We would like to give a unified framework for the very general cases, even if the general solution may be available for some specific instances (e.g., the general solution of $\partial u / \partial x_1 = 0$ is $u=f(x_2,\dots,x_d)$ where $f$ can be parameterized by a neural network). Hence, to derive the general solution, we introduce the extra fields to reformulate the BCs as linear equations (see Eq. (9)). However, Eq. (7) and (9) are equivalent if $\boldsymbol{p}_j=\nabla u_j$ holds. So we have to add additional loss terms to achieve this.
>
> Besides, as you said, we cannot directly set $\boldsymbol{p}_j=\nabla u_j$ since $(u_j, \nabla u_j)$ does not necessarily satisfy Eq. (9) (in this way, the BCs are not enforced as hard-constraints). Moreover, it is interesting to find a $\boldsymbol{p}_j$ closest to $\nabla u_j$ in the nullspace of Eq. (9). But we are not sure how this work because Eq. (7) and (9) are now not equivalent (we note that $\boldsymbol{p}_j$ and $\nabla u_j$ may be far apart).
>
> ## Q3: Richer experimental details for reimplementation.
>
> Thank you for reminding us. In addition to source codes (which are provided in the supplementary material), we have added rich experimental details to the appendix (see **Appendix A.8** in our latest version), including governing equations (PDEs as well as BCs), the derivation of ansatz for each experiment, the network architectures/hyper-parameters, choices of extended distance functions. We hope that they can help readers with reimplementation.
>
> ## Q4: Limitations of having to solve the BCs and using PINNs as a baseline.
>
> **(1)** As we discussed in Q2, the general solution is only available for a limited class of BCs. For general cases, we cannot solve the BCs analytically and can only add loss terms to encourage them to be fulfilled like vanilla PINNs.
> **(2)** In PINNs, both the PDEs and the BCs are implemented as soft constraints which serve as multiple terms in the loss function. However, as mentioned in **the second paragraph of Section 1** and in **the last paragraph of Section 2.1**, there is an unbalanced competition between the loss terms corresponding to PDEs and to BCs which can severely affect the convergence of PINNs. Even though there are methods [1] that try to balance the loss terms by adjusting the weight of each term, most of these methods are heuristic and not stable enough according to our experimental results.
>
> ## References
>
> [1] Wang, S., Teng, Y., & Perdikaris, P. (2021). Understanding and mitigating gradient flow pathologies in physics-informed neural networks. SIAM Journal on Scientific Computing, 43(5), A3055-A3081.

---

> ### Author Response · Authors · 2022-08-05
> **Looking forward to further feedback**
>
> Dear Reviewer,
>
> Thank you again for the great efforts and valuable comments. We have carefully addressed the main concerns in detail, including standard deviations and reproductivity, as suggested. We hope you might find the response satisfactory. As the discussion phase is about to close, we are very much looking forward to hearing from you about any further feedback. We will be very happy to clarify further concerns (if any).
>
> Best,
> Authors

---

> ### Author Response · Authors · 2022-08-09
> **We welcome your further comments**
>
> Dear Reviewer,
>
> Thank you very much for your efforts in reviewing our article. We think your suggestions are specific and valuable. As you can see, we have provided our answers and modifications after careful consideration:
>
> 1. We presented the results (including confidence intervals) of parallel experiments in **Appendix A.10**.
> 2. We added rich experimental details in **Appendix A.8**.
> 3. We fixed a lot of writing bugs for the presentation.
>
> Before the discussion phase ends (immediately), we strongly hope for your confirmation. Feel free to post your comments or other concerns (if any).
>
> Best, Authors

---

### Official Review · Reviewer_aRWM · 2022-07-10

**Rating:** 6
**Confidence:** 4
**Soundness:** 3 good
**Presentation:** 3 good
**Contribution:** 3 good

**Summary:**

PINN has shown potential in solving high-dimensional PDE with complex geometries. Related works generally enforce the boundary conditions as a soft constraint via boundary condition loss term, which may lead to instability and hardness in convergence. The proposed work developed a reformulation of PDE to enforce boundary conditions (Dirichlet, Neumann, and Robin BCs) as a hard constraint under the PINN framework by introducing the extra fields. Thorough experiments have been done to demonstrate the proposed approach's effectiveness in terms of accuracy (measured by mean absolute error/mean absolute percentage error).

**Questions:**

1. Why does the proposed approach enforce the boundary condition as a ***hard*** constraint? Does it require that $p=\nabla u$ is fulfilled, which is only enforced as a soft constraint?
2. How does eq11 help enforce the boundary condition and what is the motivation for using eq.10 and eq,11 as the general solution, and what's the relationship between those two equations (are the ***NN*** in both equations refer to the same thing?
3. Could more explanation be given to $g(\cdot)$ between eq.A16 and eq.A15, and their correspondences to eq.10 and eq,11.
4. What's the motivation for using extended distance functions here?

**Ethics Review Area:**

["I don’t know"]

**Limitations:**

1. It would be more reader-friendly if more connections can be built between the eq 10/11 and the derivation for each experiment shown in the supplementary.
2. It's nice to include some qualitative results and discussions on run time/convergence speed, rather than just comparing the final MAE/MAPE.



**Strengths And Weaknesses:**

***originality*** -  to the best of my knowledge, the proposed reformulation that can handle all three boundary conditions is novel in the context of PINN.

***quality*** - authors provide experiments on high dimensional/complex geometry PDEs. The derivation is mostly clear with some confusion listed in the "clarity" section and "questions" section.

***clarity*** - the motivation is clear. The derivation may be a little hard to follow when eq(10) is proposed and the extended distance function is used.  It's a bit hard to understand by boundary condition is now satisfied as a hard constraint. While numerical results have been presented, it may also be interesting to include some qualitative results to illustrate the simulation results, etc.

***significant*** - PINN is a simple approach to solving PDE, but it still with many challenges. Research works trying to develop and improve the convergence/accuracy of the PINN are of great interest.

---

> ### Author Response · Authors · 2022-08-02
> **Thanks for your valuable comments**
>
> ## Q1: Explanation for the additional equations $\boldsymbol{p}=\nabla \boldsymbol{u}$.
>
> Comparing the original PDEs (see Eq. (6) and (7)) with the reformulated PDEs (see Eq.(8) and (9)), we can find that there is a new series of additional equations $\boldsymbol{p}_j = \nabla u_j, j=1,\dots,n$ ($nd$ additional equations in total, $n$ is the dimensional of the solution $\boldsymbol{u}$ and $d$ is the spatial dimension which is also the dimension of both $\boldsymbol{p}_j$ and $\nabla u_j$). With the proposed hard-constraint framework, we can enforce the BCs (see Eq.(9)) with the price of $nd$ additional "soft constraints" which are only enforced by the loss function as you pointed out. The change in the number of "soft constraints" is $nd$ minus the number of BCs. As we mentioned in **the last paragraph in Section 3.3**, the number of BCs is far larger than $nd$ in geometrically complex systems. Therefore, the framework can reduce the total number of "soft constraints" in such systems.
>
> Besides, even if the number of BCs is lower than $nd$, we empirically find that our framework can still significantly improve the accuracy in the experiment of high-dimensional heat equation (see **Section 5.4**, where $n=1, d=10$ and the number of BCs is 2). We speculate that it may be due to the fact that competition between PDEs and BCs is greater than that between PDEs. Specifically, as we discussed in **the last paragraph in Section 2.1**, the convergence speed of PDE losses is quite different from that of BC losses. Since here we "replace" the BCs with additional PDEs $\boldsymbol{p}_j = \nabla u_j$, our framework may be beneficial to reducing the unbalanced competition.
>
> ## Q2: Explanation for Eq. (10) and (11).
>
> As you suggested, we have provided an explanation for how we construct the general solution as in Eq. (10). Besides, we also proved why Eq. (11) can satisfy the BCs and can represent the solution to the PDEs. We refer to **Appendix A.3 and A.4** (in our latest version of the paper) for further details.
>
> In addition, we would like to explain the relationship between Eq. (10) and (11) in detail. After the reformulation, we note that the BCs are converted into linear equations (see Eq. (9)). To solve a linear equation, the direct way is to find a set of basis in the null space and obtain the general solution, which is what we did in Eq. (10). By parameterization of a neural network, Eq. (10) can represent any function that satisfies the BC. Since the problem domain $\Omega$ is composed of multiple boundaries, we need to combine the general solutions of each boundary to achieve the overall approximation, which is what we did in Eq. (11). We use extended distance functions to divide and conquer, where Eq. (10) ($\mathrm{NN}_i$) is responsible for the approximation on the boundaries, while the $\mathrm{NN}$ is responsible for internal. However, we acknowledge the notation confusion between Eq. (10) and (11), which is fixed in our latest version.
>
> ## Q3: Explanation for $g(\cdot)$ in Eq. (A15) and (A16).
>
> Thanks for your suggestions. Actually, we have confused the notation $g(\cdot)$ in Eq. (10), (11) and in Eq. (A15), (A16). In Eq.(10) and (11), $g(\cdot)$ denotes the inhomogeneous term of the BC which is first given in Eq. (7). However, in Eq. (A15) and (A16), $g(\cdot)$ represents the activation function in the neural networks which is illustrated in **the second paragraph of Section 4**. To avoid this confusion, we have changed the notation of the latter to $\sigma(\cdot)$ in the latest version of our paper after the rebuttal revision.
>
> ## Q4: Motivation for using extended distance functions.
>
> The motivation is to make irrelevant items disappear from the boundary. Taking Eq. (3) as a simple example, when $\boldsymbol{x}$ is on the boundary, the irrelevant term disappears, leaving only the general solution $u^{\partial\Omega}(\boldsymbol{x})$ that satisfies the BC. If $\boldsymbol{x}$ is inside, the extended distance function is greater than 0, ensuring that the neural network $\mathrm{NN}(\boldsymbol{x};\boldsymbol{\theta})$ can exert its approximation ability.
>
> ## Q5: Reader-friendly issues.
>
> Thank you very much for your helpful feedback. We have taken your suggestions very seriously and have made the following improvements in the latest version of the paper.
> - **More connections between Eq. (10) and (11).** We have added transition content to help readers better understand the connection between Eq. (10) and (11). Please see **our modifications in Section 3.3**.
> - **Richer experimental details.** We have provided more experimental details including the derivation of the ansatz and the choices for extended distance functions in **Appendix A.8**.
>
> ## Q6: Additional qualitative results and discussions in experiments.
>
> Thanks for your constructive comments. We have added a qualitative analysis of convergence in **Appendix A.9**. We believe that this can help readers gain more insights into training PINNs.

---

> > ### Comment · Reviewer_aRWM · 2022-08-09
> > **Thanks**
> >
> > Thanks for the clarification and explanation. Just to make sure I understand response to Q1 correctly, it is still not a strictly hard constraint, i.e. the final optimized solution may not satisfy the boundary condition. The proposed method reduces the total number of soft constraints and empirically shows benefits in faster/better convergence.

---

> > > ### Author Response · Authors · 2022-08-09
> > > **Thank you for your further feedback**
> > >
> > > Thank you very much for the feedback. After the reformulation, we have three classes of equations, the reformulated PDEs, $\boldsymbol{p}=\nabla \boldsymbol{u}$, and the reformulated boundary conditions. The first two classes are equations defined inside (although $\boldsymbol{p}=\nabla \boldsymbol{u}$ is theoretically defined inside and on the boundary, we can take advantage of the generalization of the neural network without sampling points on the boundary), the last class are defined on the boundary.
> > >
> > > The ansatz we constructed is guaranteed to satisfy the reformulated boundary conditions. This leaves only the PDE terms (defined inside) in the loss function, theoretically **alleviating the unbalanced competition** (possibly due to different dimensions of definition domains of boundary conditions and PDEs) between boundary condition terms and PDE terms.

---

### Official Review · Reviewer_GzFR · 2022-07-12

**Rating:** 6
**Confidence:** 4
**Soundness:** 2 fair
**Presentation:** 3 good
**Contribution:** 3 good

**Summary:**

Physic-informed neural networks (PINNs) are a new paradigm for injecting the knowledge of governing equations into neural networks. To solve the problem, we need to consider initial conditions and boundary conditions, of which boundary conditions are hard to train for. Therefore, this paper proposes the general solutions to boundary conditions, which are later used to develop their "extra field" method. They experiment with two real-world problems.

**Questions:**

Please check the above weak points and try to answer.

**Ethics Review Area:**

["I don’t know"]

**Limitations:**

They discuss a limitation at the end of the conclusion section.

**Strengths And Weaknesses:**

This paper tackles a critical problem of PINNs since training for boundary conditions frequently collides with training for governing equations. It is needed to be balanced between them. Their method solves the problem by designing the general solutions to boundary conditions and proves several forthcoming facts.

However, this paper missed several other related works:
- https://openreview.net/forum?id=a2Gr9gNFD-J
- https://arxiv.org/abs/2012.02681

I am also not sure whether their method is theoretically correct. In general, we need to double-check whether a constructed ansatz is correct or not. Their high accuracy indirectly proves it. But it seems not sufficient.

---

> ### Author Response · Authors · 2022-08-02
> **Thanks for your valuable comments**
>
> ## Q1: Missing related works: [1] and [2].
>
> Thank you very much for reminding us of these two important related works. In our paper, we mainly discuss how to facilitate applying PINNs in geometrically complex PDEs via hard-constraint methods. The paper [1] characterized several failure modes and challenges of applying PINNs, which is one of our motivations. And the paper [2] focused on the extrapolation challenge of PINNs. We have properly cited them in Line **29** and **78** (of the latest version of our paper after the rebuttal revision).
>
> ## Q2: Theoretical correctness of the constructed ansatz.
>
> Thank you very much for your kind reminder. We agree that a rigorous theoretical underpinning is very important for our methods. We have proven how Eq. (10) becomes a general solution to the BC and why the ansatz in Eq. (11) is theoretically correct under some assumptions. We refer to **Appendix A.3** and **A.4** for detailed proof. Besides, as you mentioned, the good performance of our methods has indirectly shown that our assumptions are reasonable.
>
> ## References
> [1] Krishnapriyan, A., Gholami, A., Zhe, S., Kirby, R., & Mahoney, M. W. (2021). Characterizing possible failure modes in physics-informed neural networks. Advances in Neural Information Processing Systems, 34, 26548-26560.
>
> [2] Kim, J., Lee, K., Lee, D., Jhin, S. Y., & Park, N. (2021, May). DPM: a novel training method for physics-informed neural networks in extrapolation. In Proceedings of the AAAI Conference on Artificial Intelligence (Vol. 35, No. 9, pp. 8146-8154).

---

> > ### Comment · Reviewer_GzFR · 2022-08-09
> > **Thanks.**
> >
> > Thanks for your work. I think this paper deserves publication. The proposed method is new and has not been considered before. I think this paper adds value to the PINN community. I give +1.

---

> > > ### Author Response · Authors · 2022-08-09
> > > **Thank you for the feedback**
> > >
> > > Thank you very much for the positive feedback and appreciating our contributions!
> > >
> > > Best,
> > > Authors

---

### Official Review · Reviewer_y3Mx · 2022-07-13

**Rating:** 8
**Confidence:** 4
**Soundness:** 3 good
**Presentation:** 4 excellent
**Contribution:** 3 good

**Summary:**

This paper introduces a method for embedding the hard-constraints for boundary conditions (BCs) for physics-informed neural networks (PINNs). It introduces the "extra fields" to transform the original PDE into equivalent forms so that the BCs can become linear. This adds equations between the extra fields and the original PDE solutions, but can reduce total number of equations if the number of boundary conditions is high. The paper provides theoretical analysis of bounds for the gradients of the PDE loss terms. In experiments, the paper shows that the proposed method outperforms several baselines, and ablation study shows that it has smaller moving variance than without extra fields.

**Questions:**

N/A

**Limitations:**

The paper adequately addresses the limitations.

**Strengths And Weaknesses:**

Strengths:

The paper address the important problem in the PINN field about the dealing with complex boundary conditions. It is well-motivated, the method makes sense and is novel. The experiment evaluation is sound. The paper is written clearly. It also has good reproducibility (in terms of set up for hyperparameters and providing the code)

Weaknesses:

I don't find apparent weaknesses.

---

> ### Author Response · Authors · 2022-08-02
> **Thanks for your valuable comments**
>
> We are very grateful for your effort to review and support our work. We hope that our work can help advance the application of machine learning methods such as physics-informed neural networks (PINNs) to practical physical problems.

---

### Author Response · Authors · 2022-08-02
**Submission update for rebuttal revision**

We have submitted a new version of our paper, where modifications are listed as follows:
-  Paper (where the modified parts are highlighted in **blue**)
   -  Add transition content between Eq. (10) and Eq. (11)
   -  Add explanations for Eq. (11) and Eq. (12)
   -  Improve the presentation, including typos, confusing notations, etc
   -  Add clear descriptions for some terms
   -  Add missing related works
   -  Update the checklist
- Appendix
   - Add an explanation for the general solution (A.3)
   - Provide a theoretical guarantee for the constructed ansatz (A.4)
   - Add more experimental details (A.8)
   - Add quantitative convergence analysis (A.9)
   - Add parallel experiments as well as confidence intervals (A.10)

---

### Meta-Review · Area_Chair_mQxa · 2022-08-25

**Recommendation:** Accept
**Confidence:** Certain

**Metareview:**

The paper considers using neural networks to solve PDEs with complex geometry by incorporating hard constraint into the approximation function class. The method is interesting and useful for practical applications of neural-network based PDE solvers. The authors have adequately addressed the concerns by the referee. The meta-reviewer recommends acceptance of the paper.

**Award:**

No

---

### Decision · Program_Chairs · 2022-09-14

Accept